# Association of Postpartum Depression with Maternal Suicide: A Nationwide Population-Based Study

**DOI:** 10.3390/ijerph19095118

**Published:** 2022-04-23

**Authors:** Yi-Liang Lee, Yun Tien, Yin-Shiuan Bai, Chi-Kang Lin, Chang-Sheng Yin, Chi-Hsiang Chung, Chien-An Sun, Shi-Hao Huang, Yao-Ching Huang, Wu-Chien Chien, Chieh-Yi Kang, Gwo-Jang Wu

**Affiliations:** 1Department of Obstetrics and Gynecology, Tri-Service General Hospital, National Defense Medical Center, Taipei 11490, Taiwan; lylobgyn@gmail.com (Y.-L.L.); kung568@gmail.com (C.-K.L.); ycsobgyn@yahoo.com.tw (C.-S.Y.); 2Department of Obstetrics and Gynecology, Kang Ning Hospital, Taipei 11490, Taiwan; catebai519@gmail.com; 3Department of Psychiatry, Taoyuan Psychiatric Center, Taoyuan 33058, Taiwan; edward820828@gmail.com; 4Graduate Institute of Life Sciences, National Defense Medical Center, Taipei 11490, Taiwan; 5Department of Medical Research, Tri-Service General Hospital, National Defense Medical Center, Taipei 11490, Taiwan; g694810042@gmail.com (C.-H.C.); hklu2361@gmail.com (S.-H.H.); ph870059@gmail.com (Y.-C.H.); 6School of Public Health, National Defense Medical Center, Taipei 11490, Taiwan; 7Taiwanese Injury Prevention and Safety Promotion Association, Taipei 11490, Taiwan; 8Department of Public Health, College of Medicine, Fu-Jen Catholic University, New Taipei City 24206, Taiwan; 040866@mail.fju.edu.tw; 9Big Data Research Center, College of Medicine, Fu-Jen Catholic University, New Taipei City 24206, Taiwan; 10Department of Chemical Engineering and Biotechnology, National Taipei University of Technology (Taipei Tech), Taipei 10608, Taiwan; 11Gynecologic Oncologist Division, Department of Obstetrics & Gynecology, Chi Mei Medical Center, Tainan City 71004, Taiwan

**Keywords:** depression, postpartum, puerperal disorders, suicide

## Abstract

Background: To examine the association of postpartum depression (PPD) with maternal suicide in the Taiwanese population. Methods: We examined the medical records of women aged 18–50 years who experienced childbirth and had PPD (the study cohort, *n* = 2882), who experienced childbirth but did not have PPD (comparison cohort 1, *n* = 5764), and who neither experienced childbirth nor had PPD (comparison cohort 2, *n* = 5764) between 2000 and 2015. The patients were followed up until suicide, withdrawal from the National Health Insurance program, or 31 December 2015. Results: The rates of anxiety and depression symptoms, as well as the cumulative risk of suicide, were significantly higher in the study cohort. PPD was significantly correlated with an increased risk of maternal suicide and was associated with a greater risk of developing comorbidities such as hypertension, diabetes mellitus, hyperlipidemia, and stroke. The comparison cohorts did not differ significantly in terms of suicide risk. Conclusion: PPD was associated with a significantly higher rate of suicide and a shorter time to suicide after childbirth. Younger age, winter, and subclinical depression and anxiety positively predicted suicide in the study cohort. To prevent maternal suicide, clinicians should be observant of subclinical depression and anxiety symptoms among patients.

## 1. Introduction

Postpartum depression (PPD), a major depressive episode that occurs within 4 weeks of delivery, is one of the most common mental illnesses affecting women during and after pregnancy [1]. A meta-analysis including studies from 1989 to 2016 indicated that the prevalence of PPD ranged between 13% and 19% [2]. Studies have noted that PPD has complex pathophysiological mechanisms, from genetic factors to immune function, and rapidly fluctuating reproductive hormone levels [3]. The strong association of PPD with postpartum maternal morbidity and mortality in Western countries has been established. Patients with PPD experience symptoms including mood lability, irritability, obsessional worries, and thoughts of death [4]. Moderate-to-severe depression symptoms can persist for over 40 months after hospitalization and treatment for PPD [5].

PPD causes functional impairment and negatively affects patients’ families, especially their children [6]. The physiologically and psychologically adverse outcomes associated with PPD include preterm delivery, low birth weight, and impaired mother–infant bonding [7]. Regarding the long-term effects of PPD, the children of patients with this condition have higher rates of childhood behavioral problems and adolescent depression. Moreover, they have been documented to have poorer academic performance [8].

Reproductive hormones are pivotal to mood regulation, cognitive function, and responses to environmental stimuli. Menstrual-cycle-related changes in the levels of hormones, especially progesterone, lead to emotional disturbances in reproductive-aged women [9]. Reduced cerebrospinal fluid allopregnanolone levels have been reported in rodent research and clinical studies of depressive patients [10]. A study noted that lower progesterone levels during the postpartum period, among other changes in the levels of reproductive hormones, play a critical role in PPD [11]. An investigation reported that high-intent suicide attempts were more common when progesterone levels were low [12]. A study on Iranian women observed that lower serum progesterone concentrations were associated with a significantly higher rate of recurrent suicide attempts [13]. Despite the distinct hormonal fluctuation in the female population, other factors associated with the risk of depression have been mentioned, including age, gender, seasonality [14], and comorbid physical conditions [15]. However, their association with suicidality, especially in the postpartum population, is rarely discussed.

Up to 20% of postpartum deaths were due to suicide, and suicide during pregnancy and the postpartum period is often attempted through more lethal methods than suicide in the general female population [16]. Moreover, several cases of maternal filicide due to severe maternal depression within 12 months of delivery have been reported [17]. Thus, the prediction of and early intervention for severe PPD with high suicidality are critical concerns for clinical gynecologists and psychiatrists.

Although depression is highly prevalent worldwide, its characteristics vary across cultures. In a study by Bernert et al., depressive symptoms, especially suicide ideation, varied considerably among individuals from six European countries [18]. Further, cross-national variability in the prevalence of suicide behaviors between Western countries and Asian countries has been reported [19]. Despite the clinical importance of attempted and completed suicide among postpartum women, research on the associated or predictive factors of suicidal events among patients with PPD in the Asian population is lacking. By extracting data from medical records maintained by Taiwan’s Health and Welfare Data Science Center (HWDC), we conducted a retrospective study of women who experienced childbirth and had PPD, women who experienced childbirth but did not have PPD, and women who did not experience childbirth or have PPD. We analyzed the baseline characteristics and factors influencing suicidality among patients with PPD.

## 2. Materials and Methods

### 2.1. Data Sources

Data on 2882 women who experienced childbirth and were diagnosed as having PPD between 2000 and 2015 were extracted from Taiwan’s National Health Insurance Research Database (NHIRD). The single-payer National Health Insurance program, launched in 1995, covers up to approximately 99% of the Taiwanese population. It maintains contracts with more than 97% of local clinics, regional hospitals, and medical centers in Taiwan [20]. The NHIRD contains comprehensive information on hospital visits and clinical comorbidities, as well as anonymized information on eligibility and enrollment.

### 2.2. Ethical Approval

This study was conducted according to the Code of Ethics of the World Medical Association (Declaration of Helsinki). This study was approved by the Institutional Review Board (IRB) of the Tri-Service General Hospital (TSGH). The TSGH IRB waived the need for individual consent since all the identification data were encrypted in the NHIRD (IRB No. A202005111).

### 2.3. Study and Comparison Cohorts

We examined data on 1,936,512 women who visited the inpatient or outpatient departments of hospitals from January 2000 to December 2015. As shown in Figure 1, patients with a delivery-related discharge code (*International Classification of Diseases, Ninth Edition, Clinical Modification* (*ICD-9-CM*) codes 650–659, OP73.59, OP73.6, OP74.0-OP74.1, 81004C-81005C, 81017C-81019C, 81024C-81026C, 81028C-81029C, and 81034) or with a diagnosis of PPD (*ICD-9-CM* code 648.4) or mental disorders (*ICD-9-CM* codes 290–319) before January 2000 were excluded. Patients who completed suicide, experienced self-inflicted poisoning or injury (*ICD-9-CM* codes E950–E959) before follow-up, were aged < 18 or >50 years, received radiotherapy (*ICD-9-CM* code V58.0) or chemotherapy (*ICD-9-CM* code V58.1) before or during follow-up, or had missing data were also excluded. The study cohort comprised 2882 reproductive-aged women who experienced delivery and had PPD (*ICD-9-CM* code 648.4) for which they made over three inpatient or outpatient visits between January 2000 and the end of the follow-up period (31 December 2015). The index date was the date of the first inpatient or outpatient visit with a medical record of PPD. We followed up with the patients until the event of suicide (i.e., the outcome of interest), withdrawal from the National Health Insurance program, or 31 December 2015, whichever was the earliest. The years of follow-up (mean ± SD) of study cohort, comparison cohorts 1 and 2 were 9.24 ± 10.01, 9.27 ± 10.37, and 9.98 ± 11.53, respectively.

All patients were matched by age, socioeconomic status (indicated by insured premiums in TWD), and the season of their indexed visit to establish comparison cohorts of patients who experienced childbirth but did not have PPD (comparison cohort 1) and patients who did not experience childbirth or have PPD (comparison cohort 2). The comparison cohorts comprised 5764 patients in total. All patients were followed up through the NHIRD until the event of suicide or 31 December 2015. Definitions of the study variables are listed in Appendix A.

### 2.4. Statistical Analysis

Statistical analyses were performed using SAS software, Version 9.3, of the SAS System for Unix (SAS Institute Inc., Cary, NC, USA). Categorical variables were compared using the chi-square test for independence, whereas continuous variables were compared using the *t* test or the Fisher exact test. The cumulative risk of suicide among patients aged 18 to 50 years was estimated using Kaplan–Meier curve analysis. The significance level for all statistical analyses was *p <* 0.05.

## 3. Results

### 3.1. Clinical Characteristics

The clinical characteristics of the patients at the time of enrollment and at the end of follow-up are summarized in Appendix A, respectively. At baseline, the medical status of the study cohort for various conditions, including hypertension (HTN), diabetes mellitus (DM), hyperlipidemia, chronic obstructive pulmonary disease (COPD), chronic kidney disease (CKD), ischemic heart disease, coronary heart disease, stroke, cancer, and obesity, were modestly to significantly more favorable than were those of the comparison cohorts. However, the rates of HTN, DM, hyperlipidemia, COPD, CKD, anxiety, depression, and stroke were significantly higher *(p* < 0.001) in the study cohort at the end of follow-up than those of comparison cohort 1. Similar changes in the characteristics of the study cohort at the end of the follow-up were noted in the comparison of the study cohort with comparison cohort 2. A small proportion of the patients in comparison cohorts 1 and 2 had mood-related symptoms such as anxiety (0.31% and 0.68%, respectively) and depression (0.47% and 0.80%, respectively) (Table 1). The rates of anxiety and depression symptoms were significantly higher in the study cohort (11.55% and 42.26%, respectively).

### 3.2. Cumulative Risk of Suicide

The cumulative risk of suicide among patients aged 18 to 50 years (Figure 2) was stratified by cohort. The mean follow-up period of the study cohort was 9.24 ± 10.01 years, whereas those of comparison cohorts 1 and 2 were 9.27 ± 10.37 and 9.98 ± 11.53 years, respectively (Appendix A). The cumulative risk of suicide in the study cohort, estimated to be 20%, 38%, and 50% at the 5-year, 10-year, and 15-year follow-ups, respectively, was significantly higher than those of comparison cohort 1 (log-rank *p* < 0.001) and comparison cohort 2 (log-rank *p* < 0.001). The median durations from PPD diagnosis to suicide in the study cohort, comparison cohort 1, and comparison cohort 2 were 0.98, 5.12, and 4.26 years, respectively (Appendix A). No significant difference was observed in the cumulative risk of suicide between comparison cohorts 1 and 2 (log-rank *p* = 0.892).

### 3.3. Factors Associated with Suicide

Over the 15-year follow-up period, 313 patients (290, 13, and 10 in the study cohort, comparison cohort 1, and comparison cohort 2, respectively) completed suicide. PPD was significantly associated with an increased risk of maternal suicide. The hazard ratios (HRs) of the study cohort, with adjustment for the variables listed in Appendix A, were 19.300 (95% confidence interval (CI): 5.977–62.255) and 18.743 (95% CI: 6.667–52.689) relative to those of comparison cohort 1 and comparison cohort 2, respectively (Table 2). The other variables listed in the table were subjected to Cox regression analysis to identify the factors associated with suicide. In the comparison cohorts, the adjusted HRs were significantly lower in individuals older than 38 years. In the comparison of the study cohort and comparison cohort 1, the anxiety-symptom subgroup had a significantly higher adjusted HR of 1.353 (95% CI: 1.040–2.473, *p* = 0.034). The HRs in the depression-symptom subgroup were significantly higher than those of comparison cohort 1 (HR = 2.689, 95% CI: 1.689–4.281, *p* < 0.001) and comparison cohort 2 (HR = 2.876, 95% CI: 1.805–4.584, *p* < 0.001; Table 2). The adjusted HRs of suicide in the anxiety- and depression-symptom subgroups in all the patients were 3.053 (95% CI: 1.921–4.852) and 3.053 (95% CI: 1.921–4.852; Appendix A), respectively.

## 4. Discussion

### 4.1. Suicide and Subclinical Depression

In the comparison cohorts of women without PPD, the adjusted HRs of suicide by subclinical depressive symptoms were significantly associated with a higher risk of suicide. Suicide attempts related to subclinical depression should be taken as seriously as suicide attempts related to severe clinical depression. A study reported that young adults with mild-to-moderate depressive symptoms experienced significant suicide ideation [21]. An investigation revealed that 27% of older adults who completed suicide did not satisfy the criteria for major depressive disorder [22].

### 4.2. Suicide and Subclinical Anxiety

Significantly higher adjusted HRs of suicide were found in the women experiencing anxiety symptoms in the study cohort and in both comparison cohorts. Anxiety symptoms also affected the course and severity of depression. Compared with non-anxious depression, anxious depression was found to be associated with relatively preserved cognitive function but more severe depressive symptoms [23,24].

### 4.3. Suicide and Age

Patients older than 38 years had a significantly lower risk of suicide than patients younger than 20 years. A study reported that suicide rates varied by sex and age. Moreover, younger age and female sex were protective factors against suicide among the general population [25]. However, age and suicide in reproductive-aged women were inversely correlated. Howard et al. reported that suicidal ideation was associated with younger age, multiparity, and more severe depressive symptoms in the postpartum period [26]. A study conducted in France on female inpatients with postpartum mental illness who were jointly hospitalized with their children revealed that younger age was independently associated with a higher rate of suicide attempts [27]. Another investigation noted that cultural factors played a substantial role in the prediction of suicide attempts [28]. Overall, younger age is a significant risk factor for postpartum suicide.

### 4.4. Suicide and Season

Compared with the women without delivery, the adjusted HR of suicide was significantly higher in winter in the women with delivery and PPD (*p* = 0.023). However, a significant seasonality effect was not seen in the comparison of the women with and without PPD. In research on the use of light therapy for preventing seasonal affective disorder, a lower prevalence of winter depression in lower-latitude regions was found [29]. However, the seasonal effect on suicidality in patients with PPD has rarely been explored. A prospective study conducted in the United States suggested that seasonal variation in daylight more often increases the severity of depressive symptoms. However, the level of suicidality remained consistent regardless of this variation [30]. Although the population included in this study was from Taiwan, a-lower latitude region, the seasonal difference in suicidality was still significantly affected by delivery but not by PPD.

### 4.5. Mean Time to Suicide

Overall, the study cohort had a significantly shorter time to suicide. In addition, the mean time to suicide (in years) was slightly longer in comparison cohort 1 than in comparison cohort 2. Pregnancy and delivery, as stressful life events in either the biological or the psychosocial dimensions, might increase an individual’s vulnerability to depression. Stressful life events have been demonstrated to be related to low brain-derived neurotrophic factor levels and higher vulnerability to depression in a murine model and in human epigenetic research [31,32,33]. The appropriate management of the stress of delivery and PPD may reflect strength in the biological, psychological, and sociocultural dimensions, which decreases suicidality.

### 4.6. PDD and Physical Diseases

The proportion of patients in the study cohort who developed physical diseases was significantly higher than the corresponding proportions in comparison cohorts 1 and 2. This is notable because at baseline, the rate of physical diseases was significantly lower in the study cohort. Depression has been observed to be associated with stronger insulin resistance and higher risk of cardiac mortality [34,35]. The potential mechanisms of this association include hypothalamic–pituitary–adrenal axis dysfunction, increased proinflammatory factor activity, and reduced self-efficacy [36].

### 4.7. Limitations

This study has some limitations. First, although the medical records from the HWDC cover the majority of the Taiwanese population, they may not include suicide attempts leading to minor injuries that do not require medical support, or suicide attempts prevented by others. Second, whether the patients with PPD received adequate pharmacotherapy or psychotherapy during the follow-up period was not explored. Third, our data contained no information on depression as a product of biological or psychosocial factors or on other factors related to depression and suicide (e.g., early-life adversity, substance abuse, and lack of social support) [37].

## 5. Conclusions

PPD contributed to a significantly higher rate of suicide and a shorter time to suicide after childbirth. Younger age, the winter season, and subclinical depression and anxiety were negative predictive factors associated with suicide in individuals with PPD. PPD was associated with a greater risk of physical comorbidities, such as DM, HTN, hyperlipidemia, and stroke, at the end of the follow-up. To prevent suicide among the PPD population, clinicians should be observant of symptoms of subclinical depression and anxiety among their patients. The routine screening of PPD and the close monitoring of patients with this condition might be required for the detection of suicidality and for early coordination with mental health services.

## Figures and Tables

**Figure 1 ijerph-19-05118-f001:**
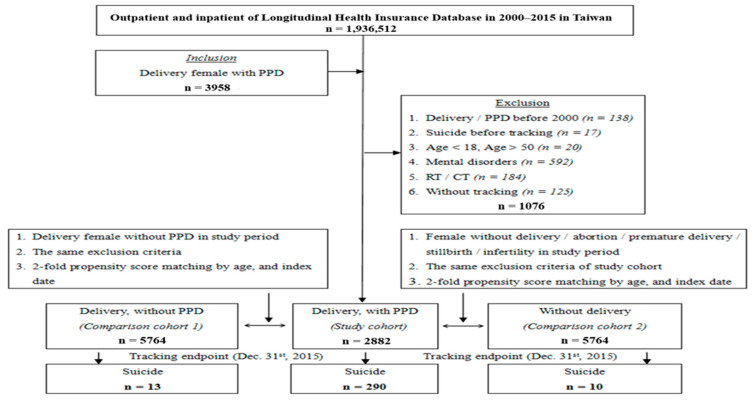
The flowchart of study sample selection. Abbreviations: PPD, postpartum depression; RT, radiotherapy; CT, chemotherapy.

**Figure 2 ijerph-19-05118-f002:**
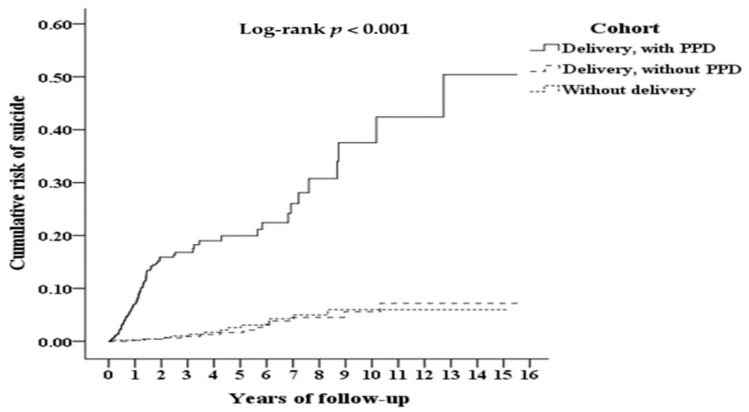
Kaplan–Meier for cumulative risk of suicide among females aged 18–50 stratified by different cohorts with log-rank test. Delivery with PPD vs. delivery without PPD: Log-rank *p* < 0.001. Delivery with PPD vs. without delivery: Log-rank *p* < 0.001. Delivery without PPD vs. without delivery: Log-rank *p* = 0.892. Abbreviation: PPD, postpartum depression.

**Table 1 ijerph-19-05118-t001:** Characteristics of study at the endpoint of follow-up.

Cohort	Delivery with PPD		Delivery without PPD			Without Delivery		
Variables	*n*	%	*n*	%	*p*	*n*	%	*p*
Total	2882	33.33	5764	66.67		5764	66.67	
**Suicide**					<0.001			<0.001
Without	2592	89.94	5751	99.77		5754	99.83	
With	290	10.06	13	0.23		10	0.17	
**Age (mean ± SD, years)**	31.25 ± 9.76		33.02 ± 7.28		<0.001	33.07 ± 7.39		<0.001
**Age groups (years)**					<0.001			<0.001
≤20	131	4.55	296	5.14		293	5.08	
21–30	1622	56.28	3202	55.55		3223	55.92	
31–34	458	15.89	1141	19.80		1198	20.78	
35–37	244	8.47	514	8.92		373	6.47	
38–40	197	6.84	257	4.46		281	4.88	
41–43	126	4.37	102	1.77		98	1.70	
≥44	104	3.61	252	4.37		298	5.17	
**Insured premium (NT$)**					0.991			0.979
<18,000	2527	87.68	5052	87.65		5060	87.79	
18,000–34,999	247	8.57	498	8.64		493	8.55	
≥35,000	108	3.75	214	3.71		211	3.66	
**HTN**					<0.001			<0.001
Without	2678	92.92	5669	98.35		5653	98.07	
With	204	7.08	95	1.65		111	1.93	
**DM**					<0.001			<0.001
Without	2714	94.17	5641	97.87		5657	98.14	
With	168	5.83	123	2.13		107	1.86	
**Hyperlipidemia**					<0.001			<0.001
Without	2804	97.29	5729	99.39		5723	99.29	
With	78	2.71	35	0.61		41	0.71	
**COPD**					0.001			0.151
Without	2827	98.09	5707	99.01		5678	98.51	
With	55	1.91	57	0.99		86	1.49	
**CKD**					<0.001			0.002
Without	2866	99.44	5757	99.88		5755	99.84	
With	16	0.56	7	0.12		9	0.16	
**IHD**					0.073			0.392
Without	2813	97.61	5660	98.20		5643	97.90	
With	69	2.39	104	1.80		121	2.10	
**CHD**					0.804			0.054
Without	2877	99.83	5752	99.79		5739	99.57	
With	5	0.17	12	0.21		25	0.43	
**Stroke**					<0.001			<0.001
Without	2820	97.85	5733	99.46		5725	99.32	
With	62	2.15	31	0.54		39	0.68	
**Cancer**					0.517			0.317
Without	2803	97.26	5620	97.50		5583	96.86	
With	79	2.74	144	2.50		181	3.14	
**Anxiety**					<0.001			<0.001
Without	2549	88.45	5746	99.69		5725	99.32	
With	333	11.55	18	0.31		39	0.68	
**Depression**					<0.001			<0.001
Without	1664	57.74	5737	99.53		5718	99.20	
With	1218	42.26	27	0.47		46	0.80	
**Obesity**					0.088			0.088
Without	2872	99.65	5755	99.84		5755	99.84	
With	10	0.35	9	0.16		9	0.16	
**Season**					<0.001			0.005
Spring	705	24.46	1319	22.88		1302	22.59	
Summer	742	25.75	1408	24.43		1493	25.90	
Autumn	820	28.45	1511	26.21		1555	26.98	
Winter	615	21.34	1526	26.47		1414	24.53	
**Location**					<0.001			<0.001
Northern Taiwan	1081	37.51	2624	45.52		2658	46.11	
Middle Taiwan	767	26.61	1662	28.83		1771	30.73	
Southern Taiwan	806	27.97	1182	20.51		1012	17.56	
Eastern Taiwan	213	7.39	283	4.91		294	5.10	
Outlets islands	15	0.52	13	0.23		29	0.50	
**Urbanization level**					<0.001			<0.001
1 (The highest)	914	31.71	2180	37.82		2141	37.14	
2	1300	45.11	2522	43.75		2502	43.41	
3	247	8.57	467	8.10		537	9.32	
4 (The lowest)	421	14.61	595	10.32		584	10.13	
**Level of care**					<0.001			<0.001
Hospital center	951	33.00	2127	36.90		2158	37.44	
Regional hospital	1362	47.26	2266	39.31		2435	42.24	
Local hospital	569	19.74	1371	23.79		1171	20.32	

Abbreviations: HTN, hypertension; DM, diabetes mellitus; COPD, chronic obstructive pulmonary disease; CKD, chronic kidney disease; IHD, ischemic heart disease; CHD, coronary heart disease; *p*: Chi-square/Fisher exact test on category variables and *t*-test on continuous variables.

**Table 2 ijerph-19-05118-t002:** Factors associated with suicide by using Cox regression.

	Delivery with PPD vs. Delivery without PPD (Reference)								Delivery with PPD vs. without Delivery (Reference)							
Variables	Crude HR	95% CI	95% CI	*p*	Adjusted HR	95% CI	95% CI	*p*	Crude HR	95% CI	95% CI	*p*	Adjusted HR	95% CI	95% CI	*p*
**Cohort**																
Study cohort	22.958	8.394	62.785	<0.001	18.743	6.667	52.689	<0.001	26.697	8.421	84.638	<0.001	19.300	5.977	62.255	<0.001
Comparison cohort 1/2	Reference				Reference				Reference				Reference			
**Age groups (yrs)**																
≤20	Reference				Reference				Reference				Reference			
21–30	0.446	0.177	1.124	0.091	0.606	0.237	1.549	0.304	0.430	0.171	1.086	0.078	0.595	0.233	1.524	0.288
31–34	0.214	0.078	0.588	0.003	0.384	0.136	1.078	0.072	0.323	0.119	0.873	0.028	0.441	0.159	1.222	0.119
35–37	0.191	0.062	0.588	0.004	0.449	0.141	1.430	0.181	0.314	0.105	0.941	0.040	0.555	0.159	1.717	0.313
38–40	0.160	0.046	0.558	0.004	0.211	0.059	0.751	0.017	0.091	0.022	0.380	0.001	0.124	0.029	0.534	0.005
41–43	0.188	0.050	0.700	0.013	0.222	0.057	0.860	0.031	0.191	0.051	0.717	0.015	0.217	0.056	0.851	0.030
≥44	0.180	0.066	0.492	0.001	0.183	0.065	0.516	<0.001	0.175	0.064	0.477	0.001	0.178	0.063	0.500	<0.001
**Insured premium (NT$)**																
<18,000	Reference				Reference				Reference				Reference			
18,000–34,999	0.743	0.104	5.333	0.768	0.578	0.074	4.505	0.602	0.579	0.081	4.160	0.589	0.573	0.074	4.457	0.596
≥35,000	1.921	0.267	13.778	0.501	4.948	0.635	38.589	0.122	2.901	0.405	20.815	0.280	5.341	0.671	42.479	0.110
**HTN**																
Without	Reference				Reference				Reference				Reference			
With	0.174	0.025	1.248	0.083	0.372	0.049	2.836	0.342	0.163	0.023	1.166	0.072	0.376	0.049	2.871	0.347
**DM**																
Without	Reference				Reference				Reference				Reference			
With	0.203	0.029	1.459	0.114	0.373	0.498	2.853	0.344	0.193	0.027	1.387	0.104	0.398	0.051	3.088	0.380
**Hyperlipidemia**																
Without	Reference				Reference				Reference				Reference			
With	0.000	-	-	0.306	0.000	-	-	0.946	0.000	-	-	0.270	0.000	-	-	0.955
**COPD**																
Without	Reference				Reference				Reference				Reference			
With	0.498	0.069	3.571	0.490	0.578	0.072	4.630	0.606	0.405	0.056	2.902	0.370	0.568	0.071	4.526	0.594
**CKD**																
Without	Reference				Reference				Reference				Reference			
With	0.000	-	-	0.641	0.000	-	-	0.971	0.000	-	-	0.581	0.000	-	-	0.974
**IHD**																
Without	Reference				Reference				Reference				Reference			
With	0.663	0.093	4.758	0.683	1.164	0.156	8.688	0.861	0.497	0.069	3.571	0.490	1.146	0.153	8.564	0.873
**CHD**																
Without	Reference				Reference				Reference				Reference			
With	0.000	-	-	0.649	0.000	-	-	0.968	0.000	-	-	0.572	0.000	-	-	0.973
**Stroke**																
Without	Reference				Reference				Reference				Reference			
With	1.617	0.399	6.562	0.484	2.314	0.537	9.969	0.250	1.055	0.261	4.281	0.913	2.210	0.511	9.546	0.276
**Cancer**																
Without	Reference				Reference				Reference				Reference			
With	0.257	0.036	1.843	0.179	0.591	0.081	4.342	0.606	0.235	0.033	1.683	0.151	0.636	0.087	4.681	0.657
**Anxiety**																
Without	Reference				Reference				Reference				Reference			
With	3.185	1.829	5.547	<0.001	1.353	1.040	2.473	0.034	2.727	1.569	4.738	<0.001	1.337	1.004	2.443	0.047
**Depression**																
Without	Reference				Reference				Reference				Reference			
With	5.131	3.405	7.733	<0.001	2.689	1.689	4.281	<0.001	4.849	3.225	7.292	<0.001	2.876	1.805	4.584	<0.001
**Obesity**																
Without	Reference				Reference				Reference				Reference			
With	0.000	-	-	0.742	0.000	-	-	0.975	0.000	-	-	0.595	0.000	-	-	0.973
**Season**																
Spring	Reference				Reference				Reference				Reference			
Summer	0.751	0.410	1.377	0.372	0.729	0.392	1.356	0.335	0.871	0.464	1.631	0.685	0.864	0.455	1.640	0.872
Autumn	0.844	0.478	1.490	0.582	0.829	0.461	1.488	0.551	0.997	0.556	1.785	0.961	1.060	0.581	1.931	0.805
Winter	1.522	0.887	2.612	0.114	1.647	0.954	2.843	0.065	1.815	1.031	3.194	0.035	1.916	1.082	3.391	0.023
**Location**					Had collinearity with urbanization level								Had collinearity with urbanization level			
Northern Taiwan	Reference				Had collinearity with urbanization level				Reference				Had collinearity with urbanization level			
Middle Taiwan	1.391	0.861	2.245	0.159	Had collinearity with urbanization level				1.321	0.815	2.143	0.233	Had collinearity with urbanization level			
Southern Taiwan	1.025	0.603	1.740	0.878	Had collinearity with urbanization level				1.088	0.645	1.836	0.706	Had collinearity with urbanization level			
Eastern Taiwan	2.025	1.025	3.998	0.038	Had collinearity with urbanization level				1.632	0.806	3.304	0.160	Had collinearity with urbanization level			
Outlets islands	0.000	-	-	0.947	Had collinearity with urbanization level				0.000	-	-	0.944	Had collinearity with urbanization level			
**Urbanization level**																
1 (The highest)	0.494	0.266	0.922	0.030	0.569	0.276	1.132	0.114	0.471	0.252	0.882	0.021	0.543	0.270	1.087	0.090
2	0.949	0.551	1.632	0.874	0.863	0.476	1.565	0.649	0.926	0.539	1.592	0.806	0.901	0.497	1.634	0.755
3	0.361	0.121	1.075	0.070	0.388	0.129	1.159	0.093	0.261	0.076	0.888	0.033	0.295	0.086	1.014	0.054
4 (The lowest)	Reference				Reference				Reference				Reference			
**Level of care**																
Hospital center	1.582	0.866	2.891	0.123	1.663	0.860	3.218	0.119	1.377	0.764	2.481	0.265	1.569	0.822	2.996	0.157
Regional hospital	1.840	1.033	3.278	0.034	1.494	0.817	2.735	0.176	1.412	0.801	2.487	0.212	1.326	0.731	2.404	0.327
Local hospital	Reference				Reference				Reference				Reference			

Abbreviations: HTN, hypertension; DM, diabetes mellitus; COPD, chronic obstructive pulmonary disease; CKD, chronic kidney disease; IHD, ischemic heart disease; CHD, coronary heart disease; HR = hazard ratio, CI = confidence interval, Adjusted HR: Adjusted variables listed in the table.

## Data Availability

Data are available from the National Health Insurance Research Database (NHIRD) published by the Taiwan National Health Insurance (NHI) Administration. Due to legal restrictions imposed by the government of Taiwan in relation to the “Personal Information Protection Act”, data cannot be made publicly available.

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
