# Peer review of "Association of Postpartum Depression with Maternal Suicide: A Nationwide Population-Based Study"

_ijerph, 2022, doi:10.3390/ijerph19095118_

Round 1
Reviewer 1 Report
The introduction does not include a review of all the variables studied, such as seasonality, which is later mentioned in the discussion.
Tables should include a footnote explaining the acronyms used.
Point 4.1 of the discussion is out of place. This topic has not been included in the introduction or in the data analyses, so it should not be in the discussion since it is not justified by the data either.
Point 4.2 of the discussion should start first with the data extracted from the study and then compare it with the data from other previous studies, which is not done properly. The same happens in the rest of the discussion points.
The data that associate suicide and seasonality are not presented in the results section, although they are in the discussion. They should also go in the results section.
Reviewer 2 Report
Overall, this is an interesting nationwide population-based study and quite a well-written manuscript that has the potential to shed more light on the complex relationship between postpartum depression and maternal suicide. The group sample size is very large. Statistical analysis is appropriate. Generally, the introduction and discussion are suitable and wide and new literature was referenced in the appropriate context.
Some points need to be considered:
Introduction:
- There is a lack of hypotheses and good scientific reasons for them.
Statistical analysis:
- There is no information on a normal distribution of the continuous variables and the value of kurtosis and skewness. The authors should conduct a Kolmogorov-Smirnov test.
- Moreover, there is no information on effect size for group differences (e.g., for parametric test Cohen's d).
Results:
- The authors should add to Tables values of the t-test and Cohen's d effect size.
Discussion:
- The authors should add to the discussion section the clinical implications of the results and future directions.
Reviewer 3 Report
Study summary:
The present study examined the association between postpartum depression with maternal suicide based on data from the Taiwan’s National Health Insurance program by comparing women with PDD to women experiencing childbirth without PPD and women who did not experience childbirth. The main finding of the present study was that PPD was associated with a significantly higher rate of suicide and a shorter time to suicide after childbirth. I think it is a very important topic and I am glad the authors raise attention for it. I have, however, some issues that need to be addressed
Major issues:
I think the study description should be more detailed. For example, I cannot find the information if all participants were included in 2000? If not, it would be helpful if the authors could include the information who was included when.
In addition, the authors mentioned that the study groups were matched by age and socioeconomic status. I would like to see a table which proves that this matching was successful.
Generally, I think that potential differences between the groups besides PDD are a major challenge for the present study, because they might have cause the revealed effects. The authors should be more clear about these differences and how they have handled these.
Another major issue are differences in the sample sizes of the cohorts which might also have influenced the results.
I am also concerned about the informative value of the study. It does not seem to be surprising that PPD was associated with higher rates of suicide, as this seems to be a well-known fact. From my opinion, the authors should emphasize the originality of their study, besides the different cultural background. By the way, I am not convinced why associations between PPD and suicide should differ between Europe and Taiwan.
Minor parts:
Abstract:
- Please add the sample size of the study cohort
- please describe more precisely compared to which comparison cohort the significant results were revealed
Theory:
- I miss a part were the authors discuss the potential role or predisposing factors before childbirth. I think this is of special relevance, as the cohort groups seem to have differed before child birthà those factors may have influenced maternal suicide
- In line 56 : Please add a reference for the sentence “Suicide is a leading cause of mortality among patients with PPD”
- I miss a definition for the term “perinatal suicide”: until what time span after child birth do you talk about “perinatal suicide”. And weren’t also suicides included in the study that happened years after child birth? Is this still referred to as maternal suicide (which does not make sense to me). The authors should be consistent in the paper when they refer to “suicide” in general” and “maternal suicide” or “perinatal suicide” throughout the text.
- And did the authors also control for additional child births? These might have also influenced additional episodes of PPD and thereby influenced maternal suicide.
Method:
- I don’t really understand when participants were included in the study. Was everyone included in 2000?
Discussion:
- I miss a discussion of the study results themselves, as the authors directly discuss potential biological mechanisms. I think that this is also important but first I would like to read a classification of the study results as well as their fitting in pervious study results
Round 2
Reviewer 1 Report
The authors have taken into account the recommendations made. The article is now clearer and more correct, so I recommend its publication
Reviewer 3 Report
Thank you for your point-to-point response, I think the manuscript improved a lot!